# Short-Term Intake of *Theobroma grandiflorum* Juice Fermented with *Lacticaseibacillus rhamnosus* ATCC 9595 Amended the Outcome of Endotoxemia Induced by Lipopolysaccharide

**DOI:** 10.3390/nu15041059

**Published:** 2023-02-20

**Authors:** Adrielle Zagmignan, Yasmim Costa Mendes, Gabrielle Pereira Mesquita, Gabrielle Damasceno Costa dos Santos, Lucas dos Santos Silva, Amanda Caroline de Souza Sales, Simeone Júlio dos Santos Castelo Branco, Alexsander Rodrigues Carvalho Junior, José Manuel Noguera Bazán, Edinalva Rodrigues Alves, Bárbara Lima de Almeida, Anne Karoline Maiorana Santos, Wellyson da Cunha Araújo Firmo, Maria Raimunda Chagas Silva, Antônio José Cantanhede Filho, Rita de Cássia Mendonça de Miranda, Luís Cláudio Nascimento da Silva

**Affiliations:** 1Laboratório de Patogenicidade Microbiana, Universidade CEUMA, São Luís 65075-120, Brazil; 2Programa de Pós-Graduação em Gestão de Serviços e Programas de Saúde, Universidade CEUMA, São Luís 65075-120, Brazil; 3Laboratório de Microbiologia Ambiental, Universidade CEUMA, São Luís 65075-120, Brazil; 4Instituto de Ciências Biomédicas, Universidade de São Paulo, São Paulo 05508-000, Brazil; 5Laboratório de Extração e Cromatografia, Instituto Federal de Educação, Ciência e Tecnologia do Maranhão, Campus Monte Castelo, São Luís 65030-005, MA, Brazil; 6Centro de Ciências da Saúde, Campus Imperatriz, Universidade Estadual da Região Tocantina do Maranhão, Imperatriz 65900-000, MA, Brazil; 7Laboratório de Química Ambiental, Universidade CEUMA, São Luís 65075-120, Brazil

**Keywords:** fruit juice, endotoxemia, probiotic

## Abstract

Endotoxemia is a condition caused by increasing levels of lipopolysaccharide (LPS) characterized by an impaired systemic response that causes multiple organ dysfunction. *Lacticaseibacillus rhamnosus* ATCC 9595 is a strain with probiotic potential which shows immunomodulatory properties. The incorporation of this bacterium in food rich in bioactive compounds, such as cupuaçu juice (*Theobroma grandiflorum*), could result in a product with interesting health properties. This work evaluated the effects of the oral administration of cupuaçu juice fermented with *L. rhamnosus* on the outcome of LPS-induced endotoxemia in mice. C57BL/6 mice (12/group) received oral doses (100 µL) of saline solution and unfermented or fermented cupuaçu juice (10^8^ CFU/mL). After 5 days, the endotoxemia was induced by an intraperitoneal injection of LPS (10 mg/kg). The endotoxemia severity was evaluated daily using a score based on grooming behavior, mobility, presence of piloerection, and weeping eyes. After 6 h and 120 h, the mice (6/group) were euthanized for analysis of cell counts (in peritoneal lavage and serum) and organ weight. *L. rhamnosus* grew in cupuaçu juice and produced organic acids without the need for supplementation. The bacteria counts were stable in the juice during storage at 4 °C for 28 days. The fermentation with *L. rhamnosus* ATCC 9595 changed the metabolites profile of cupuaçu juice due to the biotransformation and enhancement of some compounds. In general, the administration of *L. rhamnosus*-fermented juice allowed a significant improvement in several characteristics of endotoxemic status (weight loss, hypothermia, severity index, cell migration). In addition, treatment with fermented juice significantly reduced the weight of the spleen, liver, intestine, and kidneys compared to the saline-treated endotoxemic group. Taken together, our data show that short-term intake therapy of cupuaçu juice fermented with *L. rhamnosus* ATCC 9595 can reduce systemic inflammation in an experimental model of LPS-induced endotoxemia in mice.

## 1. Introduction

Lipopolysaccharide (LPS) is a structural part of the outer membrane of Gram-negative bacteria [1] that can be recognized by Toll-like Receptor 4 (TLR-4) [2,3]. Increased levels of LPS in the bloodstream are classified as endotoxemia, which may result from bacteremia and/or metabolic dysfunctions [4,5]. Metabolic endotoxemia is closely related to changes in intestinal microbiota homeostasis, due to obesity, liver damage, or other inflammatory-related chronic conditions [4,6].

Regardless of its causes, endotoxemia is a life-threatening clinical condition characterized by an impaired systemic response that provokes the dysfunction of multiple organs. Endotoxemia is a complex issue for health systems worldwide [7]. Due to the complexity of endotoxin shock physiopathology, the development of effective therapy is still a huge challenge [8].

Some evidence has shown the strong immunomodulatory effects induced by probiotic bacteria (particularly certain *Lactobacillus* sp. Strains) in experimental models of inflammatory disorders [9,10]. These results advocate for their alternative use in the treatment of clinical conditions related to endotoxemia [11,12]. These bacteria may modulate host response due to direct stimulation of immune cells and/or modification of gut microbiota [11,13,14]. A recent study showed that bacuri juice fermented with *L. rhamnosus* ATCC 9595 prolongs the lifespan of *Tenebrio molitor* larvae infected by enteroaggregative *Escherichia coli* 042 [15]. 

*Lactobacilli* are lactic acid bacteria that are normal inhabitants of the human gut and they have been also consumed as food for a long time [16]. However, most *Lactobacillus*-derived products are derived from milk, imposing an obstacle for some individuals (such as those with lactose intolerance, allergy to milk protein, and vegans), pointing to the importance of plant-derived material for the obtention of new probiotic products [17]. In this sense, the use of juice as a vehicle for these probiotic bacteria is appearing as a suitable alternative [18,19]. Furthermore, the availability of various bioactive compounds in juices may cooperate with the probiotic strains to promote beneficial effects for consumers [20,21]. 

The juice from *Theobroma grandiflorum* (Cupuassu or Cupuaçu) is very appreciated in Brazil *in nature* or industrial preparation for the production of juices, ice creams, popsicles, jams, chocolates, sweets, and is an important source of vitamins and minerals [22]; it has antioxidant properties such as vitamins C (ascorbic acid) and E (tocopherols), flavonoids, anthocyanins, and polyphenols, in addition to saccharides (glucose, fructose, and sucrose) and minerals (such as Na, K, Ca, Mg, P, Fe, Zn, and Cu) [22,23]. This work analyzed the growth of *L. rhamnosus* in cupuaçu juice and lactic acid production. We then evaluated the effects of the short-term intake of *L. rhamnosus*-fermented juice in mice submitted to endotoxemia induced by LPS. 

## 2. Materials and Methods

### 2.1. Biological Materials

#### 2.1.1. Probiotic Strain

Samples of *L. rhamnosus* ATCC 9595 are kept refrigerated at −80 °C. For each experiment, aliquots were activated in MRS broth (De Man, Rogosa, and Sharpe). 

#### 2.1.2. Pulp Characterization

The fruits of *T. grandiflorum* were obtained in the city of São Luís (Maranhão, Brazil). A sample of the plant (branch with leaf, flower, and fruit) was sent to the Herbarium “Ático Seabra” of the Federal University of Maranhão (UFMA) for identification. The proximate composition of *T. grandiflorum* was analyzed in triplicate [24]. The next fraction (carbohydrates) was obtained by calculating the difference from the other fractions analyzed. The composition of fatty acids was analyzed by the Soxhlet method, while the determination of moisture and ash was performed as described in the physical–chemical methods for food analysis [25].

### 2.2. Fermentation

The pulp was removed and stored at −20 °C. In the first fermentation assays, the pulp sample (30 g) was dissolved in 250 mL of distilled water to reach a concentration of 120 mg/mL. The pH was adjusted to 6.0 before sterilization. In parallel, a pre-inoculum of *L. rhamnosus* ATCC 9595 was prepared in MRS broth. The bacterial cells were grown at 37 °C under agitation (120 rpm). After 24 h, bacterial suspension of *L. rhamnosus* ATCC 9595 were prepared at an optical density at 600 nm (OD_600nm_) of 1.0. Then, 1 mL of this suspension was inoculated in juice or MRS broth. The cultures were incubated with shaking at 120 rpm for 48 h.

The quantification of bacterial growth was performed by plating on MRS agar. Serial dilutions were made in phosphate-buffered saline (PBS) solution after each determined period (0, 7, 14, 21, and 28 days of refrigeration). The Petri dishes were then incubated at 37 °C for 48 h, and the colony-forming units (CFU) were expressed in CFU/mL.

### 2.3. Optimization of Cultivation Conditions

The optimization of cultivation conditions was conducted through a Central Rotational Composite Design (DCCR). Inoculum (x_1_) and pulp concentrations (x_2_) were the two selected variables for the optimization (Table 1). After each test, the following parameters were determined: (i) pH values, (ii) microbial population, (iii) the relationship between bacterial growth and pH (G/pH), and (iv) lactic acid production [26].

#### 2.3.1. Quantification of Lactic Acid Content

The quantification of lactic acid in the fermentative liquid was performed using a Shimadzu high-performance liquid chromatograph, equipped with a quaternary pump r coupled to a degassing system (DGU-20A5r). The system holds an oven to control the column temperature (set at 28 °C) and an automatic injector (20 µL injection) with a diode array detector (SPD-M20A; range 190–800 nm). An ion exchange column (300 mm × 7.8 mm × 9 µm; Aminex^®^ HPX-87H, Bio-Rad, Hercules, CA, USA) was used. The elution was conducted isocratically with a mobile phase composed of 5 mM H_2_SO_4_ and with a flow of 0.6 mL/min. The software used was LC-Solutions, manufactured by Shimadzu Corporation (Kyoto, Japan) [26]. For each test, the concentration of lactic acid in an unfermented juice with the same pulp concentration (unfermented controls) was also detected. The production of lactic acid (g/L) was determined by the difference between the concentration of lactic acid in each fermented liquid and its respective unfermented control. The fermentative conditions with the best results were selected for further assays.

#### 2.3.2. Chemical Characterization of Ethyl Phases from Unfermented and Fermented Juices

The juices from the fermentative conditions with the best results were submitted to metabolites extraction using ethyl acetate as a solvent (1:1, *v/v*). The obtained extracts were used for chemical characterization. In this assay, the solvents used were purchased from J.T. Baker (Phillipsburg, USA), with all solvents of analytical and HPLC grades. The LC-HRMS system was composed of UHPLC, 1260 Infinity II system (Agilent, Barueri, SP, Brazil), and it was equipped with a high-resolution mass spectrometer (HRMS) containing a quadrupole time-of-flight mass analyzer (QTOF, Impact HD) with an electrospray ionization (ESI) source (Bruker Daltonics, Bremen, Germany).

The characterization was performed in LC-QqToF. Briefly, 1 mg of sample was dissolved in 1 mL of methanol/water (1:1, *v/v*). Samples were filtered and 20 μL aliquots were injected into the apparatus. For the analysis, a Luna C-18-HST (2.5 µm particle size; 10 × 0.21 cm) (Waters, Milford, MA, USA) analytical column was used with a mobile phase composed of water (A) and acetonitrile (B), with 0.1% *v/v* of formic acid added to both solvents. A linear gradient of 5-to-100% B in 20 min was used at a flow rate of 0.4 mL/min, with a temperature of 40.0 °C. The ionization experiments were carried out at negative mode [M-H]^−^. The parameters used for the mass spectrometry ionization source were as follows: nebulizer, 4.0 bar; dry gas flow, 8.0 L min^−1^; dry heater temperature, 180 °C; capillary voltage, 4500 V; end plate offset, 500 V; collision cell energy, 5 eV; and full-MS scan range, *m/z* 100–1000. 

### 2.4. Animal Experimentation

#### 2.4.1. Animals

This study used male C57BL/6 mice aged 6–8 weeks and weighing 20–25 g. The animals were housed in plastic cages at room temperature (23 ± 1 °C) and submitted to a 12 h light–dark cycle. They received balanced laboratory food and water ad libitum. All in vivo procedures were conducted following the laboratory animal care standards of the CEUMA University. The protocol was approved by the Ethics Committee on the use of Animals from CEUMA University (Protocol number 68/17).

#### 2.4.2. Short-Term Administration of *L. rhamnosus*-Fermented and Unfermented Cupuaçu Juice

Before the endotoxemia induction, the animals were distributed in four experimental groups that were pre-treated as follows: (i) control group (CON; *n* = 8) and (ii) LPS group (*n* = 12) received oral doses of PBS (100 µL/mouse); (iii) LPS + CUP group (*n* = 12) received oral doses of unfermented cupuaçu juice (100 µL/mouse); (iv) LPS + Lrh-CUP group received the *L. rhamnosus*-fermented juice at 10^8^ CFU/mL (100 µL/mouse). The oral administration of each sample was performed for 5 days prior to endotoxemia induction. 

#### 2.4.3. Induction and Evaluation of LPS-Mediated Endotoxemia

Each animal received an intraperitoneal injection of LPS (10 mg/kg in saline; obtained from *E. coli* serotype O111:B4; Sigma-Aldrich) [27]. Every day, the body weight and temperature were recorded and compared with the data obtained before the LPS inoculation (baseline). The severity of endotoxemia was also evaluated daily using a score as reported by Mendes et al. [28], which is based on the observation of grooming behavior, mobility, presence of piloerection, and weeping eyes. The animals (*n* = 6/group) were euthanized after 6 h and 120 h following LPS inoculation using lethal doses of 80 mg/kg ketamine hydrochloride and 10 mg/kg xylazine hydrochloride.

#### 2.4.4. Determination of Cell Population in the Blood and Peritoneal Cavity

The blood was collected by cardiac puncture using tubes having EDTA and aliquots were reserved for analysis of the total and differential cell population. The samples were then centrifuged (2000 rpm, 4 °C, 20 min) and the plasma samples were obtained and stored at −80 °C. In parallel, the mice were submitted to laparotomy followed by the introduction of 3 mL of EDTA (1 mM in PBS) into the abdominal cavity. The peritoneal lavage fluids (PELF) were transferred to a tube and stored at −80 °C. Total and differential measurements of the cell population were also performed in PELF samples. 

The leukocytes present in each sample were counted in a Neubauer chamber under microscopy (×10) after proper dilution in *Türk* solution. The differential determination of polymorphonuclear (PMN) and mononuclear (MN) leukocytes was performed using a 100 µm hanging drop of the sample obtained by cytocentrifugation at 600 rpm for 10 min. The slides were Giemsa stained and 100 cells were counted by optical microscopy at 1000 × using an oil immersion.

### 2.5. Statistical Analysis

Data were presented as means ± standard variation (SD) or percentages. The data were analyzed using the software GraphPad Prism^®^ (version 7.0) or *Statistica*^®^. The normality of distributions was determined by the Shapiro–Wilk test, and the differences between groups were evaluated by analysis of variance (ANOVA) followed by Tukey’s multiple comparison test using the Graph Prism 6.0 software. The values were considered significant when *p* < 0.05. Correlations were determined using Pearson’s coefficient (*p*) and classified as very strong (*p* ≥ 0.9), strong (0.7 ≤ *p* ≤ 0.89), moderate (0.5 ≤ *p* ≤ 0.69), weak (0.3 ≤ *p* ≤ 0.49), and negligible (*p* ≤ 0.29) [29].

## 3. Results

### 3.1. Nutritional Composition of Cupuaçu Pulp

The composition analysis (moisture, ashes content, proteins, lipids, and carbohydrates) showed that Cupuaçu pulp has a high moisture content (84.77 ± 0.2%) and dry mass (DM) of 15.23% (Table 2). The ashes content was 3.3 ± 0.9% (21.77% of DM). In addition, the pulp had 9.45% ± 0.0 of carbohydrates (62.05% of DM), 2.53% ± 0.5 of proteins (16.62% of DM), and 0.35% ± 0.0 of lipids (2.30% of DM).

### 3.2. Growth and Production of Lactic Acid by Lacticaseibacillus rhamnosus in Cupuaçu Juice

We then analyzed the growth of *L. rhamnosus* ATCC 9595 in Cupuaçu juice (120 mg/mL). This strain was able to grow in the juice without the addition of any supplement. Further, the viability of these strains was kept after 28 days of storage at 4 °C (Figure 1A). Based on these preliminary results, we evaluated the effects of inoculum concentration (x_1_) and the pulp concentration (x_2_) in the growth and organic acids production by *L. rhamnosus* ATCC 9595. 

In all assays, *L. rhamnosus* ATCC 9595 produced organic acid (as seen by reductions in pH media) displaying G/pH ratios ranging from 1.36 to 2.45 (Table 3). The best results were seen in the conditions used in assays 4 and 6. These data were used to generate the surface response graph with a linearity coefficient (R^2^) of 0.88661 (Figure 1B). Both studied variables significantly influenced the G/pH ratios (*p* < 0.05); however, the inoculum concentration had the most important effect. This was also shown by *p* values that indicated strong (*p* = 0.74) and weak (*p* = 0.47) correlations between G/pH ratios and inoculum and pulp concentrations, respectively.

Regarding the production of lactic acid, the yields ranged from 1.32 g/L to 5.72 g/L (Table 3). The surface response showed an R^2^ value of 0.74475 (Figure 1C). In this case, the tested variables did not significantly impact lactic acid levels (*p* > 0.05). However, a moderate correlation was seen by the Pearson coefficient between inoculum concentration and lactic acid production (*p* = 0.65), while the correlation was weak for pulp concentration (*p* = 0.47). Higher levels of lactic acid were found in assays 6 and 4. We selected the conditions of assay 4 to perform the animal assays.

### 3.3. Chemical Composition

To perform the chemical characterization of fermented and unfermented juices, the samples were submitted to extraction using ethyl acetate as a solvent. It is possible to observe, by superimposing the obtained chromatograms (Figure 2), changes in the profiles of the metabolites for each sample. It indicates that some metabolites were biotransformed by *L. rhamnosus*. Furthermore, it is possible to verify a greater production of some metabolites in the fermented medium.

Based on the analysis of molecular ion peaks (*m/z*) in the obtained chromatogram and in comparison with the literature data, some metabolites differentially present in each sample were identified (Table 4). Saccharose, quinic acid, tyrosol, chlorogenic acid, and diethyl succinate were only identified in unfermented juice, while Theobromine, vanillic acid glucoside, and epicatechin-(2beta- > 5,4beta- > 6)-ent-epicatechin (or epicatechin-(2beta- > 7,4beta- > 6)-ent-epicatechin) were only detected in *L. rhamnosus*-fermented juice.

### 3.4. The Administration of L. rhamnosus-Fermented Juice Reduced the Severity of LPS-Mediated Endotoxemia

We used the median severity score (SS_median_) to evaluate the progression of the LPS-mediated endotoxemia (Figure 3A). In the first 72 h, the LPS-inoculated animals without juice treatment showed higher severity scores (SS_median_ ranging from 6.0 to 9.0) than the healthy mice (those that received only PBS) (SS_median_= 4.0). After 72 h, all the groups exhibited SS_median_ equal to 4. The short-time intake of both unfermented and *L. rhamnosus*-fermented juice showed a score significantly lower (*p* < 0.0001) than the untreated endotoxemic group in this period. Importantly, the treatment with *L. rhamnosus*-fermented juice (SS_median_= 4.0) also significantly improved the outcome of disease compared to the group treated with unfermented cupuaçu juice (SS_median_= 5.0) (*p* < 0.0001). These positive effects of fermentation are seen by the analysis of data from the calculation of the Area Under Curve (AUC) (Figure 3B).

### 3.5. The Administration of L. rhamnosus-Fermented Juice Reduced the Weight Decrease Associated with Endotoxemia Induced by LPS

The induction of endotoxemia was also confirmed by the measurement of body weight and temperature. As expected, the animals had a marked reduction in body weight (Figure 3C,D) and hypothermia (Figure 3E,F) after the LPS injection. Regarding body weight, the maximum reduction was detected after 48 h for all groups (ranging from 13.58% to 17.63%). The mice treated with *L. rhamnosus*-fermented juice showed the lowest values of body weight reduction in all evaluated periods; however, significant differences were only detected after 48 h (*p* < 0.05), 72 h, and 96 h (*p* < 0.01 for both) compared with endotoxemic animals (Figure 3C). The data from AUC analysis confirmed these beneficial effects of fermented cupuaçu juice (Figure 3D). The treatment using unfermented cupuaçu juice did not affect body weight decrease. 

The three groups exposed to endotoxemia presented higher levels of hypothermia compared with healthy animals (Figure 3E,F). However, significant differences were only detected among the control groups (LPS-untreated vs. healthy animals). Although the mouse treated with *L. rhamnosus*-fermented juice displayed lower levels of hypothermia, it was not possible to observe significant differences with the other experimental groups.

### 3.6. The Administration of L. rhamnosus-Fermented Juice Amended the Weight Reduction of Some Organs Associated with Endotoxemia Induced by LPS

The weight of some organs was also evaluated after 6 h of endotoxemia induced by LPS (Figure 4). The endotoxemia was associated with significant reductions in the weight of spleens (Figure 4A; *p* < 0.0001), liver (Figure 4B; *p* < 0.0001), gut (Figure 4C; *p* < 0.05)*,* and kidneys (Figure 4D; *p* < 0.0001). The treatment with *L. rhamnosus*-fermented juice significantly reduced the weight of the spleen (Figure 4A; *p* < 0.0001), liver (Figure 4B; *p* < 0.01), gut (Figure 4C; *p* < 0.05)*,* and kidneys (Figure 4D; *p* < 0.0001) compared with the endotoxemic group. In turn, mice treated with unfermented juice also exhibited reductions in the spleen (Figure 4A; *p* < 0.0001) and kidney weights (Figure 4D; *p* < 0.0001). Other organs (lung, brain, stomach) did not present significant differences after 6 h of endotoxemia induction. No treatment-related alteration was observed for organ weights after 120 h.

### 3.7. The Administration of L. rhamnosus-Fermented Juice Amended the Migration of Cells to the Peritoneal Cavity in Mice Submitted to Endotoxemia

The peritonitis was then evaluated by the migration of cells to the peritoneal cavity (Figure 5). As expected, the endotoxemia induction significantly increased the number of leukocytes in relation to healthy animals (3.49-fold and 5.21-fold increases for 6 h and 120 h, respectively) (Figure 5A,B). The peritoneal fluids of endotoxemic mice also exhibited higher counts of leukocytes after 6 h and 120 h of LPS injection than the other animals treated with *L. rhamnosus*-fermented or unfermented juices (Figure 5). 

After 6 h, the total number of cells in PELF were reduced 3.39-fold by the short-term use of *L. rhamnosus*-fermented juice (*p* < 0.0001) and 1.97-fold (*p* < 0.01) in mice that received unfermented juice (Figure 5A). Similar results were observed after 120 h of endotoxemia induction, with 3.45-fold and 1.79-fold reductions for groups administrated with fermented and unfermented juices, respectively (Figure 5B). At this period, the administration of *L. rhamnosus*-fermented juice significantly reduced the number of cells in the peritoneal cavity concerning the treatment with unfermented juice (*p* < 0.0001). 

Consequently, the administration of *L. rhamnosus*-fermented juice also significantly reduced the number of PMN cells in both evaluated periods compared to untreated endotoxemic mice (*p* < 0.0001; 7.57-fold and 4.68-fold after 6 h and 120 h, respectively) (Figure 5C,D). Mice that received unfermented juice also displayed lower levels of PMN cells than LPS-untreated groups, with 1.84-fold and 1.51-fold reductions after 6 h (*p* < 0.01) and 120 h, respectively. Significant differences between the number of PMN leukocytes among the groups treated with unfermented and fermented juices were only observed after 6 h of endotoxemia induction (Figure 5C; *p* < 0.05).

The groups that received fermented and unfermented juices also had lower levels of MN leukocyte migration to the peritoneal cavity after 6 h and 120 h of LPS inoculation than untreated endotoxemic animals (Figure 5E,F). Again, higher rates of inhibition were seen for *L. rhamnosus*-fermented juice (2.91-fold and 3.37-fold reductions; *p* < 0.0001) than unfermented juice (2.03-fold and 1.81-fold reductions; *p* < 0.0001). Regarding the treatment, statistical differences were only observed after 120 h (Figure 5F; *p* < 0.01).

### 3.8. The Administration of L. rhamnosus-Fermented Juice Reduced the Increase of Cells in the Blood Associated with LPS-Mediated Endotoxemia

We also evaluated the effects of fermented and unfermented cupuaçu juices on the population of cells in the blood (Figure 6). The blood of endotoxemic animals showed a higher number of circulating leukocytes than all other groups in the analyzed periods (*p* < 0.0001 for all groups) (Figure 6A,B). The reductions were higher in the mice treated with *L. rhamnosus*-fermented juice (4-fold and 3.47-fold) than those treated with unfermented juice (2.45-fold and 1.50-fold). However, significant differences among the groups that received unfermented and fermented juices were only seen after 120 h (Figure 6B; *p* < 0.0001).

The treatment with both fermented and unfermented juices significantly reduced the levels of PMN leukocytes in the blood in relation to endotoxemic mice (Figure 6C, D; *p* < 0.0001). Although the reductions in PMN cells were higher in mice treated with fermented juice (3.78-fold and 2.46-fold after 6 h and 120 h of endotoxemia induction, respectively) than in those that received unfermented juice (3.19-fold after 6 h and 1.68-fold after 120 h), no significant differences were detected between both types of treatment.

Finally, we analyzed the alterations in MN leukocytes in the blood of mice submitted to each treatment schedule. We observed higher inhibitory effects for both types of treatment after 6 h of LPS-induction, with 4.01-fold and 2.42-fold reductions for mice treated with fermented and unfermented juices, respectively (Figure 6E; *p* < 0.0001). After 120 h (Figure 6F), the mice that received *L. rhamnosus*-fermented juice had *a* lower number of MN leukocytes in the blood in relation to both endotoxemic groups (4.20-fold reduction; *p* < 0.0001) and unfermented-treated mice (*p* < 0.01). The treatment with unfermented mice resulted in a 1.43-fold reduction compared with endotoxemic mice (*p* < 0.05).

## 4. Discussion

Despite many efforts, the harmful consequences of endotoxemia remain among the leading causes of mortality [34,35]. This scenario highlights the urgent need for new alternatives to prevent and/or treat this condition [8,36]. Some probiotic strains with immunomodulatory properties are indicated as attractive candidates for the management of sepsis [11,12]. Herein, we show the beneficial effects of the short-term intake of *L. rhamnosus*-fermented cupuaçu juice in a model of septic shock.

In the first stage of this study, we evaluated whether cupuaçu juice could support the growth of *L. rhamnosus* ATCC 9595. The juice was suitable for bacteria growth and production of organic acids (such as lactic acid) without any supplementation. These results are similar to those recently reported for the fermentation of bacuri juice by *L. rhamnosus* ATCC 9595 [15]. Several studies have proposed that fruit juices are considered excellent food matrices for probiotics [18,26,37,38,39]. On the other hand, fermentation can improve the activity of the phytochemicals present in the juice [18,19].

Indeed, the fermentation with *L. rhamnosus* ATCC 9595 changed the metabolites profile of cupuaçu juice due to the biotransformation and enhancement of some compounds. For instance, the production of lactic acid was significantly increased by fermentation. The increase of lactic acid content has been reported after the fermentation of bacuri juice by *L. rhamnosus* ATCC 9595 [15] and passion fruit by *L. rhamnosus* ATCC 7469 [26]. We have also showed that some compounds were specifically found in cupuaçu juice fermented by *L. rhamnosus* ATCC 9595. The biotransformative properties have been described for lactic acid bacteria during the fermentation of blueberry [18] and jujube juices [19].

We then evaluated whether the administration of cupuaçu fermented juice could improve the outcome of endotoxemic shock. Animal models that mimic the metabolic alterations seen in endotoxemic individuals are typically used for the preclinical evaluation of new therapeutic candidates [1], even though all these models have advantages and disadvantages regarding translational applications [8,40]. An example of an experimental model commonly used for mimicking endotoxemic shock is the peritoneal administration of LPS, resulting in an exaggerated inflammatory response seen in endotoxin shock, followed by dysregulation of several organs [41,42].

Our results showed that both unfermented and fermented cupuaçu juice were effective in reducing septic shock, due to their anti-inflammatory effects. Some constituents previously reported in cupuaçu juice (such as ascorbic acid (vitamin C), tocopherols (vitamin E), flavonoids, anthocyanins, and polyphenols) have antioxidant, immunomodulatory, anti-angiogenic, and anti-proliferative activities [22,23,43] Among the tested groups, the higher effects were observed for mice treated by *L. rhamnosus*-fermented juice. These effects may be related to compounds produced during the fermentation and the beneficial properties of *L. rhamnosus* ATCC 9595. Theobromine, one of the compounds specifically found in fermented juice, is a caffeine derivative which is also reported as an anti-inflammatory agent [44,45]. 

Furthermore, some *L. rhamnosus* strains have shown the ability to modulate the host immune system in different clinical situations [46,47]. The probiotic strains can also adhere to mucous membranes and protect against microbial infections that could be related to LPS-induced dysbiosis [48,49]. Additionally, *L. rhamnosus* strains can relieve hypersensitivity reactions and intestinal inflammation [50,51], and are also indicated as an adjuvant in cases of neoplasms, eczema, diarrhea, lactose intolerance, intestinal inflammation, and infections of the vaginal and urinary tracts [52,53,54].

Concerning *L. rhamnosus* ATCC 9595, this strain increased the survival time of *Galleria mellonella* larvae infected with *Candida albicans*. This action was related to the inhibition of fungal virulence factors and modulation of the immune system, as seen by the recruitment of hemocytes for hemolymph [55]. Moreover, exopolysaccharides (EPS) produced by *L. rhamnosus* ATCC 9595 were able to exhibit immunosuppressive activity when tested on peritoneal macrophages of mice stimulated with LPS. The treatment with EPS induced the production of elevated levels of IL-10 and reduced the secretion of TNF-α [56]. 

Taken together, the data obtained in this study showed that cupuaçu juice is a suitable vehicle for *L. rhamnosus* ATCC 9595, which showed high viability in the optimized conditions. During the fermentation, *L. rhamnosus* ATCC 9595 produced lactic acid without the need of any supplementation. Indeed, the fermentation changed the metabolites profile of cupuaçu juice due to the biotransformation and enhancement of some compounds. Moreover, the short-term administration of *L. rhamnosus*-fermented cupuaçu juice reduced the systemic inflammation induced by LPS in mice. These results may help in the development of probiotic therapies for the treatment and prevention of sepsis.

## Figures and Tables

**Figure 1 nutrients-15-01059-f001:**
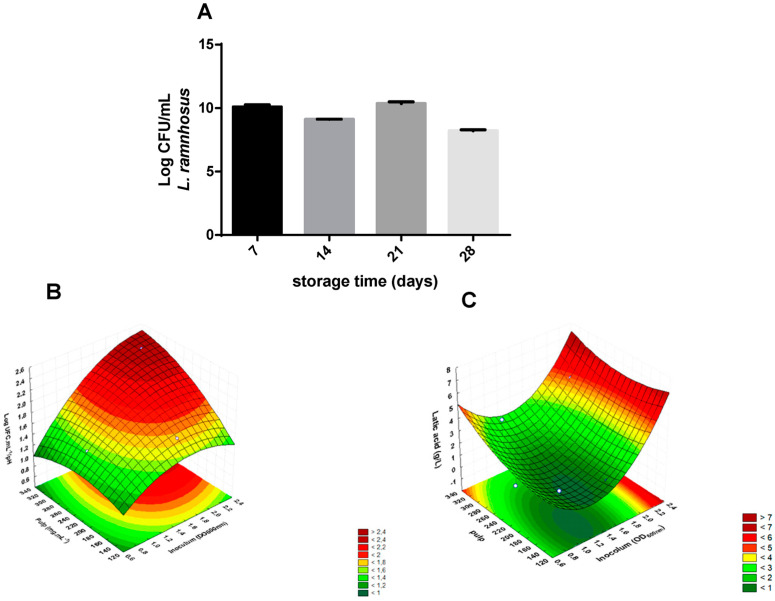
Growth and production of lactic acid by *Lacticaseibacillus rhamnosus* in cupuaçu juice. (A) Survival of *L. rhamnosus* ATCC 9595 in the fermentations of cupuaçu juice by *L. rhamnosus* ATCC 9595 during the storage period. (**B**) Response surface obtained for RVpH as a function of pulp and inoculum concentrations in cupuaçu juice fermentations by *L. rhamnosus* ATCC 9595. (**C**) Response surface obtained for lactic acid concentration as a function of pulp and inoculum concentrations in cupuaçu juice fermentations by *L. rhamnosus* ATCC 9595.

**Figure 2 nutrients-15-01059-f002:**
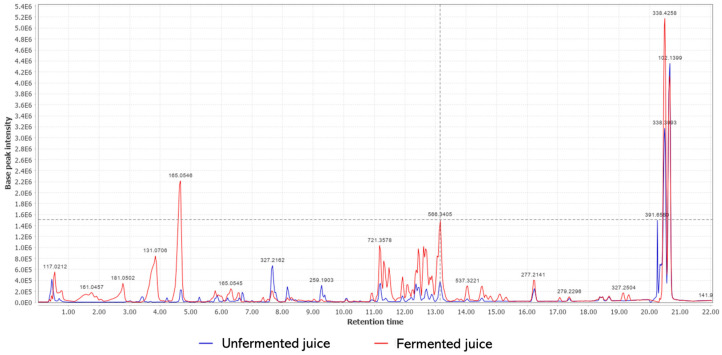
Comparative chromatograms of the ethyl acetate phase of cupuaçu juice (*Theobroma grandiflorum*) with and without fermentation by *Lacticaseibacillus rhamnosus* ATCC 9595.

**Figure 3 nutrients-15-01059-f003:**
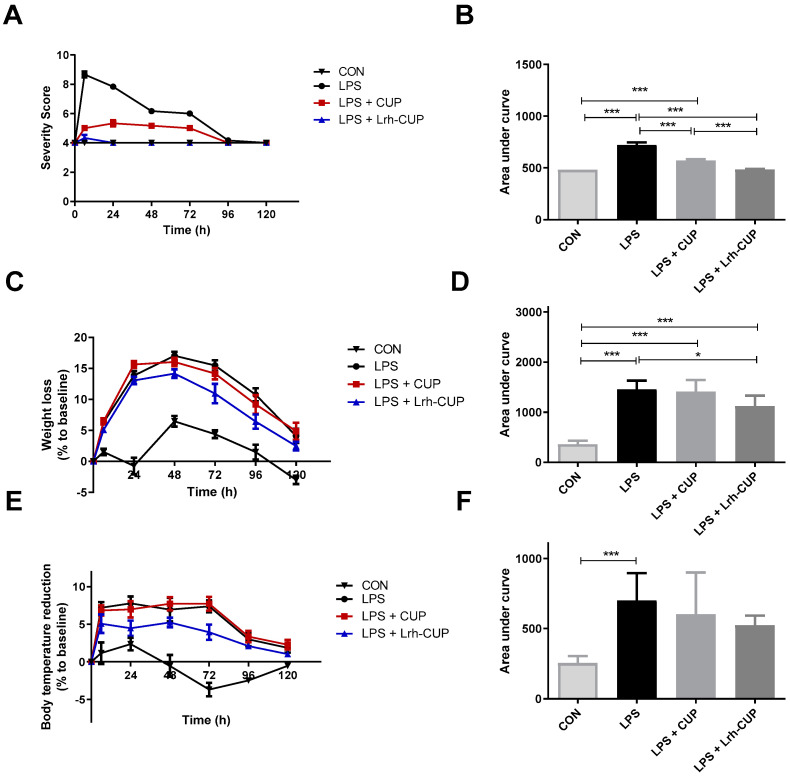
Effects of fermented and unfermented *Theobroma grandiflorum* juice on some pathologic parameters associated with LPS-mediated endotoxemia. (**A**) Kinects of variation of severity score; (**B**) Area under curve from data of severity score analysis; (**C**) Kinects of variation of weight loss; (**D**) Area under curve from data of weight loss analysis; (**E**) Kinects of variation of body temperature reduction; (**F**) Area under curve from data of body temperature reduction analysis. * Significant differences with *p* < 0.05; *** Significant differences with *p* < 0.0001. LPS: mice submitted to LPS-mediated endotoxemia; LPS + CUP: mice treated with unfermented *T. grandiflorum* juice and submitted to LPS-mediated endotoxemia; LPS + Lrh-CUP: mice treated with *L. rhamnosus*-fermented *T. grandiflorum* juice and submitted to LPS-mediated endotoxemia.

**Figure 4 nutrients-15-01059-f004:**
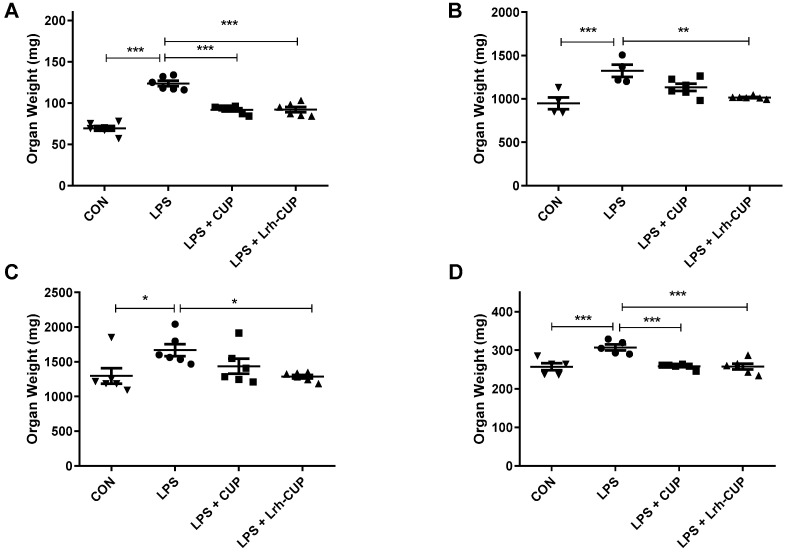
Effects of fermented and unfermented *Theobroma grandiflorum* juice on the weight of some organs of mice submitted to LPS-mediated endotoxemia. (**A**) Spleen; (**B**) Liver; (**C**) Gut; (**D**) Kidneys. * Significant differences with *p* < 0.05; ** Significant differences with *p* < 0.01; *** Significant differences with *p* < 0.0001. LPS: mice submitted to LPS-mediated endotoxemia; LPS + CUP: mice treated with unfermented *T. grandiflorum* juice and submitted to LPS-mediated endotoxemia; LPS + Lrh-CUP: mice treated with *L. rhamnosus*-fermented *T. grandiflorum* juice and submitted to LPS-mediated endotoxemia.

**Figure 5 nutrients-15-01059-f005:**
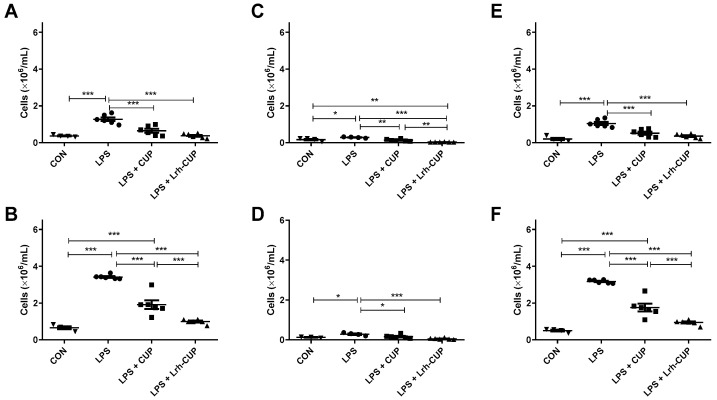
Effects of fermented and unfermented *Theobroma grandiflorum* juice in the migration of cells to the peritoneal cavity in mice submitted to LPS-mediated endotoxemia. (**A**) Total leukocytes in the peritoneal cavity after 6 h of LPS-mediated endotoxemia; (**B**) Polymorphonuclear leukocytes in the peritoneal cavity after 6 h of LPS-mediated endotoxemia; (**C**) Mononuclear cells in the peritoneal cavity after 6 h of LPS-mediated endotoxemia; (**D**) Total leukocytes in the peritoneal cavity after 120 h of LPS-mediated endotoxemia; (**E**) Polymorphonuclear leukocytes in the peritoneal cavity after 120 h of LPS-mediated endotoxemia; (**F**) Mononuclear cells in the peritoneal cavity after 120 h of LPS-mediated endotoxemia. * Significant differences with *p* < 0.05; ** Significant differences with *p* < 0.01; *** Significant differences with *p* < 0.0001. LPS: mice submitted to LPS-mediated endotoxemia; LPS + CUP: mice treated with unfermented *T. grandiflorum* juice and submitted to LPS-mediated endotoxemia; LPS + Lrh-CUP: mice treated with *L. rhamnosus*-fermented *T. grandiflorum* juice and submitted to LPS-mediated endotoxemia.

**Figure 6 nutrients-15-01059-f006:**
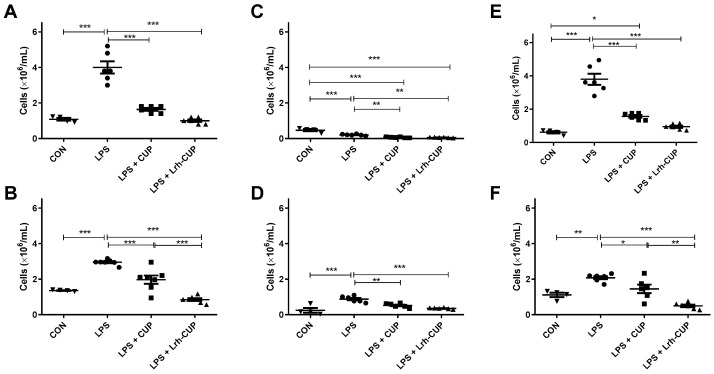
Effects of fermented and unfermented *Theobroma grandiflorum* juice in leukocytes population in the blood of mice submitted to LPS-mediated endotoxemia. (**A**) Total leukocytes in the peritoneal cavity after 6 h of LPS-mediated endotoxemia; (**B**) Polymorphonuclear leukocytes in the peritoneal cavity after 6 h of LPS-mediated endotoxemia; (**C**) Mononuclear cells in the peritoneal cavity after 6 h of LPS-mediated endotoxemia; (**D**) Total leukocytes in the peritoneal cavity after 120 h of LPS-mediated endotoxemia; (**E**) Polymorphonuclear leukocytes in the peritoneal cavity after 120 h of LPS-mediated endotoxemia; (**F**) Mononuclear cells in the peritoneal cavity after 120 h of LPS-mediated endotoxemia. * Significant differences with *p* < 0.05; ** Significant differences with *p* < 0.01; *** Significant differences with *p* < 0.0001. LPS: mice submitted to LPS-mediated endotoxemia; LPS + CUP: mice treated with unfermented *T. grandiflorum* juice and submitted to LPS-mediated endotoxemia; LPS + Lrh-CUP: mice treated with *L. rhamnosus*-fermented *T. grandiflorum* juice and submitted to LPS-mediated endotoxemia.

**Table 1 nutrients-15-01059-t001:** Coded and decoded levels of independent variables.

Variables	−1.41	−1	0	+1	+1.41
Inoculum density (OD600)	0.77	1.00	1.55	2.10	2.33
Pulp concentration (mg/mL)	135.25	163.00	230.00	297.00	324.75

**Table 2 nutrients-15-01059-t002:** Nutritional Composition of Cupuaçu Pulp.

Attribute	Moisture	Ashes	Proteins	Lipids	Carbohydrates *
Content (%)	84.77 ± 0.2	3.3 ± 0.9	2.53 ± 0.05	0.35 ± 0.0	9.45 ± 0.0
% Of Dry Mass	-	21.77	16.62	2.30	62.05

* Calculated using the equation = 100 − (protein + lipids + ash + moisture).

**Table 3 nutrients-15-01059-t003:** Effects of inoculum density and pulp concentration on growth and lactic acid production by *L. rhamnosus* ATCC 9595 in cupuaçu juice.

Assay	OD_600nm_	Pulp (mg/mL)	Growth (CFU/mL)	Final pH	Lactic Acid (g/L)	G/pH Ratio	G/[La] Ratio
1	1.00	163.00	6.13	4.5	1.32	1.36	4.65
2	1.00	297.00	7.32	4.4	3.96	1.66	1.85
3	2.10	163.00	8.41	4.9	3.25	1.72	2.59
4	2.10	297.00	9.07	3.7	4.98	2.45	1.82
5	0.77	230.00	7.29	4.7	0.65	1.55	11.25
6	2.33	230.00	8.46	3.9	5.72	2.17	1.48
7	1.55	135.25	7.44	4	1.44	1.86	5.17
8	1.55	324.75	9.43	4.6	1.13	2.05	8.34
9	1.55	230.00	8.67	4.2	1.25	2.06	6.92
10	1.55	230.00	9.19	4.5	1.14	2.04	8.05

**Table 4 nutrients-15-01059-t004:** Identification of substances differentially present in the ethyl acetate phase of cupuaçu juice (*Theobroma grandiflorum*), with and without fermentation by *Lacticaseibacillus rhamnosus* ATCC 9595.

Sample	RT (min)	[M-H]	Proposed Substance	Reference
Unfermented	0.46	341.1159	Saccharose	[30]
	0.70	191.0227	Quinic acid	[30]
	3.03	137.0244	Tyrosol	[31]
	5.17	353.1801	Chlorogenic acid	[30]
	5.86	173.1176	Diethyl succinate	[31]
Fermented	2.78	181.0502	Theobromine	[32]
	8.16	329.2318	Vanillic acid glucoside	[30]
	13.84	633.3425	Epicatechin-(2beta- > 5,4beta- > 6)-ent-epicatechin Epicatechin-(2beta- > 7,4beta- > 6)-ent-epicatechin	[33]

RT: retention time; [M-H]: negative mode.

## Data Availability

Not applicable.

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
