# Peer review of "Short-Term Intake of Theobroma grandiflorum Juice Fermented with Lacticaseibacillus rhamnosus ATCC 9595 Amended the Outcome of Endotoxemia Induced by Lipopolysaccharide"

_nutrients, 2023, doi:10.3390/nu15041059_

Round 1

Reviewer 1 Report

This manuscript investigated the effects of Theobroma grandiflorum juice fermented with Lacticaseibacillus rhamnosus ATCC 9595 on the endotoxemia induced by lipopolysaccharide, and found that the short-term intake of fermented juice could reduce systemic inflammation of lipopolysaccharide-induced endotoxemia in mice. As we all known, the function of fermented fruit juice may be related to the kinds and compositions of the main active ingredients including polyphenol, flavonoid, vitamin c and other characteristic functional component. Unfortunately, the content changes of corresponding polyphenols, flavonoid monomers and their total amounts were unknown. The other questions are listed below:

Line 25: (Theobroma grandiflorum→(Theobroma grandiflorum)

Line 26: in→on

Line 34: The temperature unit should be revised.

Line 89 and 107: I did not find Table 1 and Table 2 in the full text.

Line 94: The solid content of the juice sample used should be given.

Author Response

Dear reviewer,

Thank you so much for all suggestions that significantly improved our manuscript. Please find bellow the point-to-point response for all your comments. The changes are highlighted in yellow in the manuscript.

This manuscript investigated the effects of Theobroma grandiflorum juice fermented with Lacticaseibacillus rhamnosus ATCC 9595 on the endotoxemia induced by lipopolysaccharide, and found that the short-term intake of fermented juice could reduce systemic inflammation of lipopolysaccharide-induced endotoxemia in mice. As we all known, the function of fermented fruit juice may be related to the kinds and compositions of the main active ingredients including polyphenol, flavonoid, vitamin c and other characteristic functional component. Unfortunately, the content changes of corresponding polyphenols, flavonoid monomers and their total amounts were unknown.

Our response: Thank you for this observation. We have included the results from the chemical composition of the juice before and after the fermentation.

The other questions are listed below:

Line 25: (Theobroma grandiflorum→(Theobroma grandiflorum)

Our response: We have performed this change.

Line 26: in→on

Our response: We have performed this change.

Line 34: The temperature unit should be revised.

Our response: We have performed this change.

Line 89 and 107: I did not find Table 1 and Table 2 in the full text.

Our response: Sorry for our mistake, we have added the respective tables.

Line 94: The solid content of the juice sample used should be given.

Our response: Sorry for our mistake, we have added this information.

Reviewer 2 Report

Short-term intake of Theobroma grandiflorum juice fermented with Lacticaseibacillus rhamnosus ATCC 9595 amended the outcome of endotoxemia induced by lipopolysaccharide it is an interesting article about the potential of Theobroma grandiflorum juice as a probiotic after the addition of Lacticaseibacillus rhamnosus ATCC 9595.  However, in order to be published, the article needs major changes These are indicated below, but also in the attached document:

Row 25  - You forgot to close the bracket

R82 If the title is 2.1. Origin and maintenance of probiotic strains you should not talk about the other analyzes performed, they should be treated under a separate heading

R 97-99 – “1 mL aliquots 97 of each bacterial suspension” Were there more than one bacterial suspensions?

The bacterial suspension I assume refers to L. rhamnosus… This part should be rewritten to be a little clearer for the reader.

R 138 what the control group received?

Table 1 and 2 are attached as supplementary materials? I can't find them in the document

R  245-255 Marked words should not be italicized

Actually, there are many words in the document that should not be written in italics, as well as words that should be written in italics and are not. The entire document should be checked.

The discussion part should include comparisons of the own results obtained in this study with those existing in the literature. The discussions of this article practically iterate the properties of the tested juice components, which is not ok, considering that the tests carried out are not in this direction. The discussion part must be rewritten.

Author Response

Dear Reviewer 2,

Thank you so much for all your comments. We have changed the manuscript following them (please see the highlighted sentences in the updated manuscript). Following,  we present a point-to-point response to yours comments. 

Short-term intake of Theobroma grandiflorum juice fermented with Lacticaseibacillus rhamnosus ATCC 9595 amended the outcome of endotoxemia induced by lipopolysaccharide it is an interesting article about the potential of Theobroma grandiflorum juice as a probiotic after the addition of Lacticaseibacillus rhamnosus ATCC 9595.  However, in order to be published, the article needs major changes These are indicated below, but also in the attached document:

Row 25  - You forgot to close the bracket

 Our response: We have performed this change.

R82 If the title is 2.1. Origin and maintenance of probiotic strains you should not talk about the other analyzes performed, they should be treated under a separate heading

Our response: We have corrected this.

R 97-99 – “1 mL aliquots 97 of each bacterial suspension” Were there more than one bacterial suspensions? The bacterial suspension I assume refers to L. rhamnosus… This part should be rewritten to be a little clearer for the reader.

Our response: It was used only on bacteria. We have corrected this.

R 138 what the control group received?

Our response: We have rewritten this section to make it clearer.

Table 1 and 2 are attached as supplementary materials? I can't find them in the document

Our response: Sorry for our mistake, we have added the respective tables.

R  245-255 Marked words should not be italicized. Actually, there are many words in the document that should not be written in italics, as well as words that should be written in italics and are not. The entire document should be checked.

Our response: We have corrected this.

The discussion part should include comparisons of the own results obtained in this study with those existing in the literature. The discussions of this article practically iterate the properties of the tested juice components, which is not ok, considering that the tests carried out are not in this direction. The discussion part must be rewritten.

Our response: Thank for this comment. As we have included new results, the discussion section was updated accordingly.

Round 2

Reviewer 1 Report

All the comments have been answered, and the relevant text and data were supplemented. The physicochemical components and characteristic components before and after fermentation have been given. I therefore recommend minor revision.

Line 114: There is an extra slash.

Line 119: and (iv).

Line 143: There is an extra symbol ("]"), please check.

Line 145: ml→ mL.

Line 374: Figure 5F.

Line 428: due to.

Author Response

Dear reviewer 1,

Thank you so much for your comments. We have modified the manuscript following your instructions. The changes are highlighted in yellow in our updated manuscript. 

Best regards.

Reviewer 2 Report

The author made improvements to the article and I believe that it can be published in the present form.

Author Response

Dear reviewer 2,

Thank you so much for your comments that improved our manuscript.

Best regards.